# Development of the “Applied Proteomics” Concept for Biotechnology Applications in Microalgae: Example of the Proteome Data in *Nannochloropsis gaditana*

**DOI:** 10.3390/md20010038

**Published:** 2021-12-29

**Authors:** Rafael Carrasco-Reinado, María Bermudez-Sauco, Almudena Escobar-Niño, Jesús M. Cantoral, Francisco Javier Fernández-Acero

**Affiliations:** Microbiology Laboratory, Institute of Viticulture and Agri-Food Research (IVAGRO), Marine and Environmental Sciences Faculty, University of Cadiz (UCA), 11500 Puerto Real, Spain; rafael.carrasco@uca.es (R.C.-R.); maria.bermusau@alum.uca.es (M.B.-S.); almudena.escobar@uca.es (A.E.-N.); jesusmanuel.cantoral@uca.es (J.M.C.)

**Keywords:** applied proteomics, biotechnology, industrial, microalgae, omics

## Abstract

Most of the marine ecosystems on our planet are still unknown. Among these ecosystems, microalgae act as a baseline due to their role as primary producers. The estimated millions of species of these microorganisms represent an almost infinite source of potentially active biocomponents offering unlimited biotechnology applications. This review considers current research in microalgae using the “omics” approach, which today is probably the most important biotechnology tool. These techniques enable us to obtain a large volume of data from a single experiment. The specific focus of this review is proteomics as a technique capable of generating a large volume of interesting information in a single proteomics assay, and particularly the concept of applied proteomics. As an example, this concept has been applied to the study of *Nannochloropsis gaditana*, in which proteomics data generated are transformed into information of high commercial value by identifying proteins with direct applications in the biomedical and agri-food fields, such as the protein designated UCA01 which presents antitumor activity, obtained from *N. gaditana*.

## 1. Introduction

In the middle of the 20th century, an event took place that would bring about a fundamental change in understanding how a living organism functions: the application of new molecular biology techniques and methodologies. The development of “omics” techniques (genomics, transcriptomics, metabolomics, and proteomics) that allow, from a single experiment, the generation a large amount of information has opened the doors to a new postgenomic era [1].

The deciphering of the human genome showed to the scientific community that the main interest was not only in understanding the information present at molecular levels (RNA, proteins, metabolites, etc.), but also the interactions between them [2]. From the techniques developed from “omics” technologies, it is possible to analyse the information represented by the different elements of molecular biology, and thus obtain a large amount of relevant data in a short time from a single experiment (with the corresponding replicates for every condition analysed) [2].

Genomics is dedicated to the study of the genome, that is, the complete set of the DNA sequence in an organism. Following the logic of molecular biology, genomics leads next to transcriptomics, in which the RNA present in an organism is studied [3,4]. The translation of this RNA gives rise to proteins, the fundamental objective of proteomics, in which we study the set of all the proteins and post-translational modifications expressed in a cell, tissue, or organ at a given time and under specific environmental conditions [5,6].

Proteomics offers the capability to identify and quantify all the set of proteins expressed by a microorganism at a determined moment. This large volume of information allows us to understand the biological behaviour of that microorganism at a given time, under given environmental conditions [1]. Proteomics can therefore be defined as the science that studies the proteome, which is the set of proteins present in a cell, a tissue, or a biological organism. These proteins, detected in proteomics assays, are the result of the expression of the thousands of genes and their variations (mutations) that form DNA, the processing of RNA, and the post-translational modifications that the proteins undergo [7]. Understanding the proteome in its entirety is one of the greatest challenges facing modern biology, and represents a major advance, since proteins are responsible for controlling biological functions. In fact, it is currently the objective of the Human Proteome Project (HPP) promoted by the Human Proteome Organisation (HUPO). Proteome assays have shown the low correlation between RNA expression and proteins expression [1].

Proteomics provides complementary information to genomics and transcriptomics and enables a better understanding of complex biochemical processes at the molecular level [8]. In addition, not all mRNAs are translated into proteins under the same regulation, and one polypeptide chain may possess many different forms through differential splicing of the mRNA and post-translational modifications. Consequently, a vast amount of information may remain missing unless proteomics analyses are performed [8], since these are the most relevant analyses [9].

Microalgae biomass is in great demand for many industrial applications due to the potential application of microalgae for contributing to the solution of many problems caused by human activities, for example, greenhouse gas emission, water contamination, fossil fuel depletion, and the novel therapies for various diseases that need to be developed [10,11]. Presently, microalgae are being widely cultivated using a variety of different processes. This interest has situated microalgae at the main axis of efforts to develop new biotechnological tools supported by the “Blue Growth” initiative of the European Union (EU) [12]. “Blue Growth” is the EU’s long-term strategy for supporting sustainable growth in the marine and maritime sectors. Oceans, seas, and rivers have long been drivers of the European economy, offering great potential for innovation and growth; they represent the maritime contribution towards achieving the overall goals of the Europe 2020 strategy for smart, sustainable, and inclusive growth (“Commission of the European Union: “Blue Growth” opportunities for marine and maritime sustainable growth”). In Europe, the “blue” economy is estimated to account for approximately 5.4 million jobs and the generation of gross added value of around EUR 500 billion a year. The Blue Growth strategy represents one of the most ambitious plans of humanity, a plan in which microalgae have an important role. A major subobjective is to secure the implantation of the “circular economy” model [13]. At this point, better understanding of the biotechnological potential of microalgae is crucial for discovering new compounds with biological activity; this potential is very considerable, especially since it is known that there are several million different marine species, compared with a total of around 250,000 species of terrestrial plants [1,14].

Proteomics approaches to the study of microalgae may help to unravel their behaviour, and this information will be very useful for increasing the commercial production and profitability of microalgae cultures [1]. More detailed understanding will be needed to determine the best microalgae culture conditions, oriented towards faster growth and the increased production of some of the more interesting fractions [8]. A specific challenge, and at the same time an opportunity, for applying proteomics approaches in microalgae research is in sample conditioning as the proteomic profile of a microalga varies depending on many parameters. The conditions affecting microalgae growth include the effect of a growth medium on the synthesis of the bioproduct of interest [15], nutrient deprivation [16,17], and the effect of a particular mode of cultivation [18,19].

By applying the proteomics approach to the microorganism *Nannochloropsis gaditana*, it has been possible to identify various proteins with potential industrial application. In fact, more than four hundred different proteins with industrial applications have been identified [20]. The identification of proteins with industrial application from a single proteomics assay is a recent development in classic proteomics [20], from which the concept of “applied proteomics” has been introduced [21]. This concept is based on the combination of the potential of microalgae with the corresponding proteomics dataset. A novel bio-algorithm has been developed by means of which proteomics data generated are combined with available patents databases, thus allowing the comparison of the proteins identified in a particular proteomics study with those sequences registered in any patent document, revealing native proteins with direct industrial applications, following different parameters [21].

This review is focused on unravelling details of the proteomics approach to microalgae with special emphasis on the development of the applied proteomics concept in *N. gaditana*. These processes have enabled potential new industrial applications to be found, ranging from new proteins with potential capacity against phytopathogenic fungi, to new antitumoral compounds.

## 2. “Omics” Applications in Microalgae

The “omics” techniques have revolutionised science and its applications. The combination of genomics, metabolomics, and proteomics are currently the spearhead of biotechnology, allowing procedures and discoveries that were impossible a few years ago [22]. The “omics” techniques are based on the research and study of genes, proteins, and metabolic reactions, as well as on the interactions between them. Digital information is growing rapidly, in terms of the so-called Five V’s (volume, velocity, veracity, variety, and value). Hence, this is hailed as the “big data era” [23,24,25]. The big data generated by “omics” research (such as genetics, proteomics, and metabolomics) has now become more widely available for all researchers [26,27].

Genetic engineering for the selection of traits of interest in microalgae has made great progress in the last decade [11,20,22]. Of course, within the advances generated in recent years in the field of molecular biology, special relevance has been gained by the sequencing techniques that allow all the genetic material of an organism to be obtained very rapidly.

Recognized as one of the most important scientific advances ever, the polymerase chain reaction (PCR) is a very fast way of creating unlimited copies of DNA from an original strand. In just a few hours, the amount of DNA needed for analysis can be obtained. Thus, from a small sample, DNA can be amplified, making it possible to detect diseases or contaminants in a product. It has become a fundamental tool for detecting the SARS-CoV-2 virus, proving once again that PCR is one of the most important advances in the history of science.

Metabolic engineering has generally become a central strategy for optimizing genetic and biosynthetic pathways within cells to increase the yield and rate of any metabolite [28]. Most successful biosynthetic advances are reported for the model eukaryotic algal systems *Chlamydomonas* and *Chlorella* but also with prokaryotic microalgae and cyanobacteria which have the capacity to produce high-value biocompounds such as proteins, through the expression of certain genes in algal cells, or by using genetic knockout and knockdown strategies to change the metabolic flow to a specific path [28].

Genetic manipulation is a field of study in constant evolution. In microalgae, important advances have been reported, such as the expression of transgenes, using riboswitches, as a novel mechanism for gene regulation in algae [22]; inducible nuclear promoters and luciferase reporter genes; and inducible chloroplast gene expression [1]. RNA interference (RNAi) technology can also be used to downregulate the expression of particular genes. Current applications of gene editing, novel platform designs for proteins, and computational modelling will also be helpful toward increasing production of metabolites of interest from microalgae [22].

## 3. Proteomics

Proteomics is one of the most highly developed and widely applied “omics” in biotechnology [1]. The term “proteomics” was coined in the early 1990s, when the methodology was improved to the point of obtaining reproducible results and being able to identify proteins with great precision [29,30]. The main advance was the development of identification techniques using mass spectrometry, known as MALDI (matrix-assisted laser desorption/ionization) and ESI (electrospray ionization), that allow the soft ionization of a peptide useful for protein identification [29,31]. Currently, this sophisticated equipment enables samples to be identified without the prior need to isolate the protein in a gel. Hence, gel-based proteomics assays have been relegated to specific objectives [6,29]. The basic strategy followed in proteomics assays is based on extracting the protein component of the species in question, processing the proteins extracted to obtain the peptides that make them up, separating these peptides using chromatographic techniques, and identifying them using high-resolution mass spectrometry (Figure 1) [20].

In the context of molecular techniques, it is essential to know the proteome of the microorganism and to have the facilities to produce “something” (a biological product) of industrial interest. This information is obtained from proteomics assays. Proteomics reveals what is responsible for the behaviour of the organism (its phenotype) [32]. The proteo-me of the organism presents a wide range of information, which can be transformed into high-value data for many fields of investigation. With the information from proteomics research, the best strategies in genetic engineering can be designed [33]. Supported by proteomics studies, the best techniques for the production of proteins and metabolites can be understood; the recent applications of gene editing, novel platform designs for proteins, and computational modelling, all supported by proteomics studies, will also contribute toward increasing production [34].

The most efficient strategy to obtain information on how microalgae function is proteomics: it essentially gives us information on the “software” of microalgae [32]. A microalga changes its expression proteins depending on changes in environmental conditions (i.e., culture conditions) because the various proteins expressed are seen to be different [35]. A proteomics assay provides the opportunity to determine the behaviour of the microalgae and the relationships of that behaviour with the environment [21].

Despite the development of proteomics platforms, not many approaches have been developed to study microalgae proteomes yet, since microalgae are not model organisms [16,32,36,37,38,39,40,41]. The main aim of most of the proteomics assays reported has been to improve biodiesel production, as well as other relevant aspects of the conditions for their possible culture. These studies have used a great variety of proteomics strategies, for both protein extraction methods, and techniques for the analysis of the proteins extracted.

Microalgae cell walls vary widely in composition among species and may contain glycoproteins, cellulose, hemicellulose, proteins, uronic acid, xylan, mannose, various minerals, and layers of algaenan [41]. *Euglena* spp. have a proteinaceous cell pellicle and diatoms have a hard cell wall with pores [42,43]. Dinoflagellates possess high endogenous levels of nucleic acids, polysaccharides, pigments, salts, and other components that could interfere with subsequent analysis. Choice of protein extraction technique is thus dependent largely on the microalga under study.

The methods usually applied for the disruption of microalgal cells for protein extraction preparation involve the application of mechanical, chemical, and electrical forces. The method of extraction selected for a protein will depend on the type of cell wall possessed by the microalga. Examples include species such as *Nannocholoropsis oculata*; with this microalga different disruption methods have been studied. The method with the best ratio of protein extraction is high-pressure homogenization, under the following conditions: 125 MPa, 1–6 passes, 0.8 g L^−1^. Another microalga studied for different protein extraction methods is *Chlorella vulgaris*; with this microalga the process studied with the best ratio for protein extraction is high-pressure homogenization, but under conditions different from those for *Euglena gracilis* [15].

Conversely, dinoflagellates have a different cell structure. The method usually reported to give the best protein extraction level is mechanical homogenization [20,44,45]; however, in *Dunaliella salina*, for example, freezing and thawing has been described [46]. In *N. gaditana*, disruption by mechanical homogenization presents satisfactory results, as reported in the research of Fernandez-Acero et al. (2018) [47]. In *C. reinhardtii*, enzyme digestion is described [48]. Among these various cell disruption methods, high-pressure homogenization has been shown to be the most effective method in most microalgae with rigid cell walls [49].

Having considered the methods for protein extraction, our review moves on to the various different methods used for the analysis of the proteins obtained. These methods include matrix-assisted laser desorption/ionization time-of-flight/time-of-flight (MALDI TOF/TOF) analysis, qualitative and quantitative liquid chromatography with tandem mass spectrometry (LC-MS/MS) approaches, and sodium dodecyl sulphate–polyacrylamide gel electrophoresis (SDS–PAGE GEL).

The SDS–PAGE gel is a gel for electrophoresis and offers two possibilities: analysis of protein expression in one or two dimensions (1DE or 2DE). The SDS–PAGE 1DE allows protein expression to be studied only by molecular weight; this method is limited, as it is only possible to analyse a minimal number of proteins and is normally only used when the investigation is focused on one or a few groups of proteins and their expression [22]. The SDS–PAGE 2DE incorporates another variable with the second dimension—the isoelectric point. The method is based on the displacement of molecules in a pH gradient. Amphoteric molecules, like amino acids, separate in a medium in which there is a difference in potential and a pH gradient [23]. The SDS–PAGE 2DE method has the advantage of making it possible to differentiate proteins having the same molecular weight but with a different isoelectric point [23].

The second-generation proteomics innovation was the incorporation of a prior stage of high-performance liquid chromatography (HPLC) to tandem mass spectrometry (MS/MS); this allows the proteins to be separated in accordance with different parameters. One of the parameters most frequently used is by their molecular weight, using MS/MS.

Two methods exist for protein identification: one is the soft ionization method for macromolecules, using electrospray ionization (ESI), and the second is matrix-assisted laser desorption/ionization (MALDI) [8]. The second-generation proteomics techniques made it possible to analyse all the proteins present in an organism, but the protein expression could not be quantified [20]. The proteins analysed by second-generation proteomics start to be quantified with isobaric tags for relative and absolute quantitation (iTRAQ), which leads to third-generation proteomics.

The proteomics analysis method used depends on the type of study being carried out. For example, the methods that have been used with *Euglena gracilis, Nannochlorpsis oceanica*, *Chromochloris zoofingiensis,* and *Chlorella pyrenoidosa* are SDS–PAGE; and nano-LC–MS/MS. SDS–PAGE has also been used with other microalgae such as *Phaeodactylum tricornutum* together with an assay performed using MALDI–TOF–TOF–MS. In *Chlorella spp*. the proteomics assay has been performed using 2DE and OFFGEL fractionation, nano-LC–MS/MS and iTRAQ LC–MS/MS [8]. *Dunaiella salina* has been analysed by proteomics techniques using 2DE and electro spray ionization quadrupole time of flight tandem mass spectrometry (ESI–QTOF–MS/MS). *Nannochloropsis oceanica* has been subjected to proteomics analysis using PAGE and LC–ESI–MS/MS. *Tetraselmis subcordiformis* has been assayed using SCX and 2D nano-LC–MS/MS [8,19,50,51]. In proteomics assays it is essential to select the correct method of extraction and the correct tools for analysing the results depending on the research interest.

These efforts will provide valuable information for our understanding of the biology of this important group of microorganisms and will strengthen the impact of their use in biotechnological applications [44]. This approach will lead in the future to the design of novel microalgae, selecting the most favourable characteristics and eliminating the undesirable ones in order to optimise the production of the molecule of interest required by the industrial sector in question [1].

Proteomics assays represent the large-scale study of the function and structure of proteins. The meaning of “proteome” refers to the set of proteins encoded by the genome, as well as the added post-translational modifications [51]. It could be said that DNA is the blueprint for life, and proteins are the tools that make living machines work. DNA is like the “hardware” and proteins the “software” of an organism [1]. The proteome is more variable in contrast with the more stable genome [51]. Furthermore, the genetic code has four nucleotides, whereas proteins are built from 20 different amino acids, post-translational modifications and the chemical constituents of the proteins–phosphates, sugars, fats, and other associated proteins. In addition, the proteins have several isoforms, are involved, and mixed in different metabolic pathways and usually they join to form complexes made up of multiple proteins [51,52]. The set of proteins produced also depends on cell type, cell shape, and cell function; the tissue in which the cell resides; the ways in which the environment affects them; and the stage of development the cell has reached. This high degree of complexity present in proteomics assays means that proteomics studies will be the most relevant level of analysis for microorganisms [9]. In addition, the study of the proteome in non-model organisms will help to unravel their behaviour and pathways and will provide much valuable information about them [1,21,31].

## 4. The Applied Proteomics Concept and Its Application in the Proteome of Microalgae

Proteomics allows us to understand how an organism works at a given time, comparing two different situations (the set of proteins and their levels of expression). For example, polyunsaturated fatty acids (PUFAs), which are one of the products of greatest industrial interest available from microalgae, are synthesised in greater amounts under certain stress conditions. This difference in synthesis of PUFAs is because the pool of proteins expressed under stress conditions is different compared with under non-stress conditions [1]. The key proteins in specific metabolic pathways for the accumulation of PUFAs are found within this pool of proteins, and can be identified in a proteomic assay, [1,21].

The applied proteomics concept has transformed proteomics studies. Proteomics has always been used as an “omics” tool for generating a large volume and wide range of information for resolving complex biological questions (see Figure 2). Applied proteomics, however, can identify proteins with potential industrial applications in a single proteome experiment with the corresponding repetitions for each condition analysed (see Figure 3) [20,21]. In a recent study, the sequences of 500 peptides obtained from *Tetradesmus obliquus* and selected with the PeptideRanker algorithm were used; 25 antioxidant and angiotensin-converting enzymes were found; then, 4 of these 25 peptides were synthesized and the same activities were confirmed by in vitro testing. With this finding, a virtual screening can now be performed to identify compounds of interest [53,54]. The increasing availability of software programs, tools, and databases promises a wide range of drug discoveries in the design strategy for many specific therapeutic targets [54].

The wide field of proteomics has revolutionised the discovery of natural products by means of a set of tools and technologies [30,53,55] that allow natural products with potential industrial applications to be explored, using the applied proteomics concept [20,56]. The applied proteomics concept is able to identify proteins with potential industrial application from a single proteomics assay. This concept in conjunction with the biological potential of microalgae, can be used to discover many proteins with industrial application in microalgae.

A good example is the case of an applied proteomics assay with *N. gaditana*, where more than 400 proteins with potential industrial applications have been identified (Figure 3). Applied proteomics has transformed the proteomics data of *N. gaditana* into a high value industrial information [21]. Two of these proteins with potential applications have been studied further: (1) one “hypothetical protein” (GenBank: XM_005854224.1) that was over expressed in atomized samples; and (2) the other “resistance to phytophthora” (GenBank: XM_005852605.1) [57]. The hypothetical protein belongs to the prohibitins family, which is a protein family that may be used as treatment of diseases related to proliferative disorders [58]. The second protein, “resistance to phytophthora”, is capable of conferring oomycete resistance, particularly to *Phytophthora infestans* [58,59]. To confirm the existence of these proteins and check their availability from *N. gaditana* as a potential target for biotechnology product development, an RT-PCR was used. The result of the PCR amplification of the anti-phytophthora and prohibitin genes confirmed that they are detected in *N. gaditana* [20].

Synthesis of the recombinant protein from the “hypothetical protein” (GenBank: XM_005854224.1) was carried out to check the biological activity of this protein and prove that the concept of “applied proteomics” works [21]. This recombinant protein was named UCA01; this protein (patent number: 201930775) has been synthesized in a strain of *E. coli*, Rosetta gami 1, a specific strain for expressing proteins, and in the TOP10 strain as plasmid construction storage. Total RNA from *N. gaditana* was isolated, and the complementary DNA (cDNA) synthesis amplification was performed to prevent introns and alternative splicing. The induction of UCA01 protein expression has been performed in Rosseta gami 1 (DE3), transformed with the vector pET28Luci containing the cDNA of UCA01. The induction of protein expression was performed with isopropyl-β-d-1-thiogalactopyranoside (IPTG). Having purified the protein UCA01, in order to evaluate its biological activity (antiproliferative activity) the purified protein was tested against the following cell lines: human colorectal adenocarcinoma epithelial cell line, Caco-2 (ATCC^®^ HTB-5 37); human hepatocellular carcinoma cell line, HepG2 (ATCC^®^-HB-8065); and human endothelial cell line, a non-tumour cell line EA.hy926 (ATCC^®^ CRL-2922™) as control. The cell lines were cultivated in the presence of different concentrations of UCA01. After the incubation period, cell viability was evaluated

This analysis showed that UCA01 inhibited the proliferation and cell viability of two human cancer cell lines, the liver cancer cell line HepG2 and the colon cancer cell line Caco-2, thus demonstrating the antiproliferative activity of the *N. gaditana* recombinant protein UCA01 [21]. In addition, the antiproliferative effect in Caco-2 presented a trend of positive correlation with the concentration of recombinant UCA01. Moreover, *N. gaditana* UCA01 did not show antiproliferative effect against the non-tumour cell line EA.hy926. This result suggests that this protein has a selective effect, acting only on these tumour cell lines, but not against non-tumour cells.

Analysing the *N. gaditana* proteome with the newly developed bioinformatics algorithm has allowed the identification of *N. gaditana* proteins with potential industrial applications [60]. The algorithm is based on sequence homology, semantic homology, potential domains annotations, and comparison with other data sources such as United States patents [21]. This investigation has opened a new approach to the possibilities of proteomic studies, leading to the development of the new concept of applied proteomics.

Applied proteomics has transformed the proteome of a microalga into data of high industrial value. Analysing the proteome of *N. gaditana* has enabled the identification of 488 proteins with potential industrial applications. Among them, the biological activity of the selected protein (UCA01) has been demonstrated against two human tumour lines (Caco-2 and HepG2), where the UCA01 protein showed antiproliferative activity against both tumour cell lines.

This study has demonstrated, for the first time, the potential antitumor activity of the recombinant prohibitin protein UCA01 derived from *N. gaditana*, and the suggested antiproliferative activity of a type-2 prohibitin (PHB2). This represents the synthesis of a new approach with great potential against tumour diseases, based on the new concept of applied proteomics. The use of non-model organisms represents a potential source of new proteins, with potential industrial application, which in turn could play an important role in some of the fields described. While model organisms have perfectly described behaviour and components, non-model organisms may offer new sources of characteristics with potential industrial applications. [1,21]. Applied proteomics is, in fact, a tool able to identify proteins with potential industrial application from all the many proteins identified in a proteomics assay by means of a bioinformatic algorithm [21]. The difference between standard proteomics and applied proteomics is that the latter has the advantage of enabling proteins with potential industrial application to be identified once the proteomics assay has been performed [20].

## 5. Conclusions

Microalgae have been positioned as a key element with the capacity to help solve major problems facing humanity, including the greenhouse effect, the development of new drugs, the production of superfoods, and environmental protection. Given the potential applications of microalgae and their property as an eco-friendly product for the environment (due to their capacity to fix atmospheric CO_2_), microalgae are at the centre of EU research policies and programs. Currently, microalgae are a source of new bioproducts for biomedicine and agri-food, as well as the basis for the development of new biotechnological applications.

“Omics” techniques have revolutionised the science of microbiology and its applications, and the combination of “omics” tools are currently the spearhead of biotechnology, enabling procedures and discoveries that were impossible a few years ago.

Proteomics is a fundamental technique for unravelling the molecular dynamics involved in all biological processes given the functional information provided by the proteome. This review has shown a new approach to the possibilities of “omics” tools, with the focus on proteomic studies and especially on the new concept of applied proteomics. As an example, analysing the *N. gaditana* proteome using a newly developed bioinformatics algorithm has allowed the identification of many proteins in *N. gaditana* that offer potential industrial applications.

The concept of “applied proteomics” has been developed, showing that the data obtained may be transformed from a “simple” protein identification to a product or process patent. The proteomics approach has been positioned as a useful tool for developing novel biotechnology applications.

The case of *N. gaditana*, where more than 400 proteins with direct industrial applications have been identified, was successfully analysed thanks to the development of a bioinformatics algorithm described by Carrasco et al. (2021). A technique has been created that has allowed the development of a recombinant protein (UCA01) as an anti-tumour treatment, which has been successfully tested on tumour cell lines in the liver and colon (Cells line: HepG2. Caco-2) (patent number: 201930775). The applied proteomic concept has been created as a means of obtaining direct industrial applications from proteomics data.

## Figures and Tables

**Figure 1 marinedrugs-20-00038-f001:**
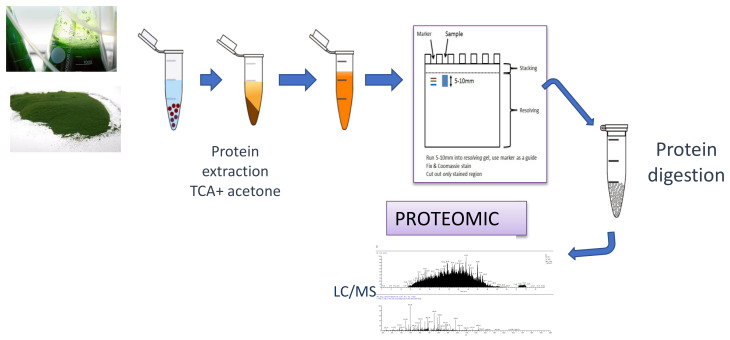
Schematic workflow of a proteomics assay, from *N. gaditana* protein extraction to protein identification by MS/MS.

**Figure 2 marinedrugs-20-00038-f002:**
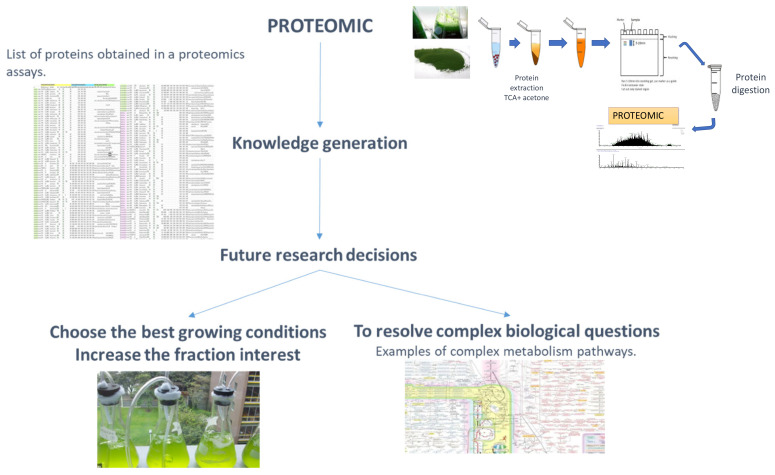
Schematic workflow of proteomics information obtained and some of its possible uses for basic research and biotechnology. (1) Proteomics: with corresponding workflow for the extraction of proteins. (2) Knowledge generation, with a representation of a list of proteins obtained in a particular proteomics assay, after MS/MS analysis. (3) Future research decisions. (4) Choosing the best growth conditions, to increase the fraction of interest, with an image of a microalgae culture. (5) For resolving complex biological questions, with examples of complex metabolism pathways.

**Figure 3 marinedrugs-20-00038-f003:**
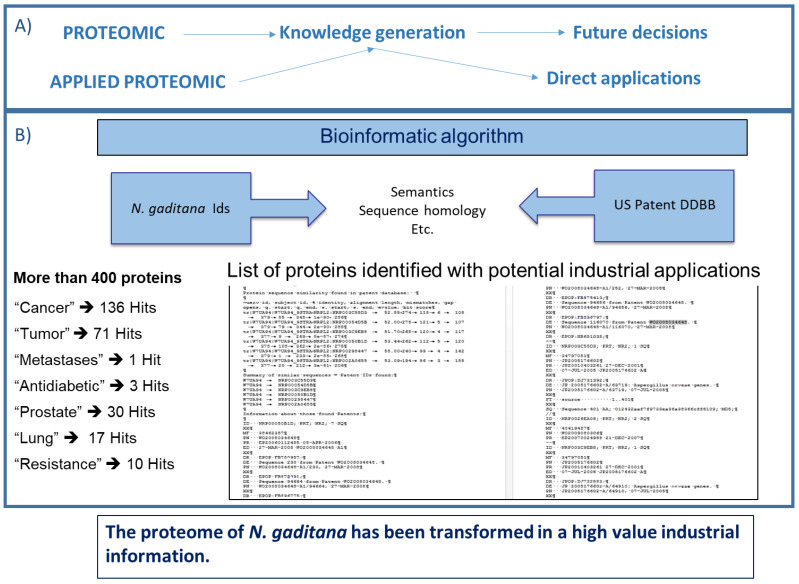
(**A**) The applied proteomics concept for transforming the proteome data obtained into potential real-world applications. (**B**) A bioinformatic algorithm of the applied proteomics concept enables the identification of more than 400 proteins in *N. gaditana* with potential industrial application, and the number of search “hits” for different keywords.

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
