# Peer review of "Development of the “Applied Proteomics” Concept for Biotechnology Applications in Microalgae: Example of the Proteome Data in Nannochloropsis gaditana"

_marinedrugs, 2021, doi:10.3390/md20010038_

Round 1

Reviewer 1 Report

The review of Carasco-Reinado et al. addresses a very interesting and important topic of current algal biotechnology, the use of proteomics to analyze cell behavior at certain conditions. The topic is of potential interest to the algal biotechnologist. Yet, currently, it suffers from several shortcomings.

  • The text has a very varied quality including English spelling and grammar. This makes the review hard to comprehend at places and should be improved and unified.
  • There is a very variable amount of details in different parts. The introduction and early parts sometimes read more like an advertisement than a critical review. Whilst the last part on applied proteomics reads as a project report including unnecessary methodological details. For suggestions, how this can be improved see below.
  • The early parts are very general, at places even superficial and lack sufficient details although these would be highly interesting to the reader and would make the later text easier to understand. There should be a critical and specific overview of the topic including the limitations and potential followed by several examples of use. This is so far true only for the last section, which, however, is too detailed.

I would recommend to add a comprehensive paragraph explaining clearly what are the limitations and advantages of proteomics for the algal study and specifically what are its advantages compared to other “omics” techniques. This is several times touched upon in the MS but the account is never comprehensive and it remains superficial and not very informative. For example, it should be mentioned if proteomics is dependent on genome sequence as to major extent are the other omics techniques, if it can be used on all organisms or only on the established models, etc. Furthermore, I would add a section which would specifically introduce the methodology of proteomics from the sample preparation to protein isolation and proteomics analyses including the details and principles, potential limitations and ways how these can be overcome. This would be of potential interest to the readers particularly from the biotechnology field. Furthermore, the main part of the review is the last part concerning the applied proteomics. This is potentially interesting but the advantages of the technique for the field of biotechnology and its differences from the standard techniques should be more explained and stressed.

Minor points:

  • Unify the use of genus names and abbreviations. Currently, there is a mix of E. gracilis and Euglenias (should be Euglena) gracilis within single paragraph. The typos in the organism´s names should be corrected, e. g. Nannochloropsis is mentioned several times and almost every time a new typo is introduced.
  • Explain the abbreviations at their first appearance and not randomly.
  • Some parts of the text are oversimplified leading to misunderstandings, e. g. l. 129-131 where gene knock-out and knock-down strategies are implied for both Chlamydomonas and Chlorella. The CRISPR/Cas9 system is yet to be established to reliable use in Chlamydomonas reinhardtii and it was so far never used in Also, examples of knock-down in Chlorella are rare if existent. Similarly, later on the same section eukaryotic algae such as Chlamydomonas reinhardtii are mentioned but examples are presented from unrelated prokaryotic cyanobacteria. This is largely misleading.
  • Throughout the MS, it is mentioned several times that a single set of samples is used for proteomics and it is claimed as the main advantage of the method. I am not sure what is meant by this statement? Firstly, to understand any process in any organism several replicates at the same conditions are always analyzed. Is this meant as the single set of samples? Secondly, in principle for any technique a “single set of samples” containing samples from different replicates is containing the information so in this sense proteomics is no different than other techniques.

Author Response

Reviewer 1:

Comments and Suggestions for Authors

The review of Carrasco-Reinado et al. addresses a very interesting and important topic of current algal biotechnology, the use of proteomics to analyze cell behavior at certain conditions. The topic is of potential interest to the algal biotechnologist. Yet, currently, it suffers from several shortcomings.

The text has a very varied quality including English spelling and grammar. This makes the review hard to comprehend at places and should be improved and unified.

  • Following the advice of this Reviewer, a native English speaker has revised the manuscript.

There is a very variable amount of details in different parts. The introduction and early parts sometimes read more like an advertisement than a critical review. Whilst the last part on applied proteomics reads as a project report including unnecessary methodological details. For suggestions, how this can be improved see below. The early parts are very general, at places even superficial and lack sufficient details although these would be highly interesting to the reader and would make the later text easier to understand. There should be a critical and specific overview of the topic including the limitations and potential followed by several examples of use. This is so far true only for the last section, which, however, is too detailed.

  • Following the advice of Reviewer, the early parts have been completed with more and detailed information, and critical comparison with methods usually used with microalgae. Some of the specific details in the last section have also been deleted.

I would recommend to add a comprehensive paragraph explaining clearly what are the limitations and advantages of proteomics for the algal study and specifically what are its advantages compared to other “omics” techniques. This is several times touched upon in the MS but the account is never comprehensive and it remains superficial and not very informative. For example, it should be mentioned if proteomics is dependent on genome sequence as to major extent are the other omics techniques, if it can be used on all organisms or only on the established models, etc.

  • Following the advice of the Reviewer, the paragraph explaining clearly what are the limitations and advantages of proteomics for the study of algae, and specifically what are its advantages compared to other “omics” techniques, has been included in the main text. In addition the paragraph includes explanations of whether proteomics is dependent on the genome sequence, as are the other “omics” techniques, to a major extent, and whether it can be used on all organisms or only on the established models.

Furthermore, I would add a section which would specifically introduce the methodology of proteomics from the sample preparation to protein isolation and proteomics analyses including the details and principles, potential limitations and ways how these can be overcome. This would be of potential interest to the readers particularly from the biotechnology field. Furthermore, the main part of the review is the last part concerning the applied proteomics. This is potentially interesting but the advantages of the technique for the field of biotechnology and its differences from the standard techniques should be more explained and stressed.

  • Following the advice of the Reviewer, a section has been added introducing the methodology and alternatives in sample preparation and isolation, and describing proteomics analyses including the details and the principal potential limitations. The advantages of the technique for the field of biotechnology and its differences compared with standard techniques have been explained more fully and given more emphasis.

Minor points:

Unify the use of genus names and abbreviations. Currently, there is a mix of E. gracilis and Euglenias (should be Euglena) gracilis within single paragraph. The typos in the organism´s names should be corrected, e. g. Nannochloropsis is mentioned several times and almost every time a new typo is introduced.

  • Following the advice of the Reviewer, the use of genus names and abbreviations has been corrected.

Explain the abbreviations at their first appearance and not randomly.

  • Following the advice of the Reviewer, an explanation of the abbreviation is always included when it first appears.

Some parts of the text are oversimplified leading to misunderstandings, e. g. l. 129-131 where gene knock-out and knock-down strategies are implied for both Chlamydomonas and Chlorella. The CRISPR/Cas9 system is yet to be established to reliable use in Chlamydomonas reinhardtii and it was so far never used. Also, examples of knock-down in Chlorella are rare if existent. Similarly, later on in the same section eukaryotic algae such as Chlamydomonas reinhardtii are mentioned but examples are presented from unrelated prokaryotic cyanobacteria. This is largely misleading.

  • Following the instructions of Reviewers 1 and 2 the references about the CRIPR/Cas9 System have been deleted in the main text, because it is considered outside the topic.

Throughout the MS, it is mentioned several times that a single set of samples is used for proteomics and it is claimed as the main advantage of the method. I am not sure what is meant by this statement? Firstly, to understand any process in any organism several replicates at the same conditions are always analyzed. Is this meant as the single set of samples?

  • We thank the Reviewer for the suggestion. The expression “a single set of samples” has been replacement by “a single experiment” and we this explain properly; several replicates under the same conditions are always analyzed in the experiment.

Secondly, in principle for any technique a “single set of samples” containing samples from different replicates is containing the information so in this sense proteomics is no different than other techniques.

  • We thank the Reviewer for this question. Proteomics differs from the other “Omics” in the type information it provides, which describes the behavior of a microorganism at a particular moment and under determined environmental conditions. Applied proteomics is different because, in comparison with other “Omics”, it does not provide information for understand the microorganism or organism; rather, applied proteomics identifies the proteins revealed in a proteome assay (with three replicates) and matches them with potential industrial applications. In the case of N. gaditana, the proteome study revealed the protein information about this microalga in under different conditions. Then the applied proteomics concept, using the bioinformatic algorithm, enabled the identification of those proteins with potential industrial applications. This last part represents the novelty of the work: the proteins identified that have potential industrial applications are found from a single proteome experiment (with the necessary replicates).

Reviewer 2 Report

The idea of « Applied Proteomics » in microalgae biotechnology sounds interesting and worth reviewing. However, I struggled going through the text due to clarity issues. I would recommend an extensive round of english grammar and spelling editing, preferentially by a native speaker, to make up for this important clarity issue. Please also check for typos (e.g. in the title it should be « through » and not « thought »). Additionally, I have a few more specific comments listed hereafter that I hope will help the authors to improve the quality of their draft :

More specific comments :

- Very often, proteomic approaches lead to the identification of only a few peptides per protein with limited coverage. In most microalgae, which also have uncomplete genome annotations (if any), identifying proteins with confidence thus represents a real challenge. Wouldn’t that represent a major drawback for the proposed applied proteomics approach ?

- Lines 133-143 : This part about CRISPR/Cas9 genome editing is nice and all, but isn’t it a little bit out of topic ? Or at least, genome editing applications in the « applied proteomics » context should be more explicitely explained : could the CRISPR/Cas9 system help overproducing proteins of interest, delete some potentially problematic proteases ?....

- Line 254 : « The applied proteomics have been transformed the proteomics data of N. gaditana in a high value industrial information [21]. ». This is the key part and yet, the underlying concept is not explained at all. How does the algorithm identifying proteins of biotechnological interest functions? Is it purely based on sequence alignments, on potential domains annotations ? On GO terms, biased towards biomedical applications ? And if in the Nannochloropsis dataset more than 400 proteins with potential industrial applications could be found, how to narrow down the list to candidates of interest ? Natural abundance should be a criterion I imagine.

- Lines 266-291 : This part denotes in the middle of the review… It looks like a mixture of Materials and Methods with a little bit of a Results section. Since the reported cloning, recombinant expression and antitumoral tests were previously reported in another reference [21] (Carrasco-Reinado R. et al.), the main essential finding that UCA01 was identified through the « applied proteomics » pipeline and has antitumoral activity should be reported in a more condensed paragraph.

Author Response

Reviewer 2

The idea of « Applied Proteomics » in microalgae biotechnology sounds interesting and worth reviewing. However, I struggled going through the text due to clarity issues. I would recommend an extensive round of English grammar and spelling editing, preferentially by a native speaker, to make up for this important clarity issue. Please also check for typos (e.g. in the title it should be « through » and not « thought »). Additionally, I have a few more specific comments listed hereafter that I hope will help the authors to improve the quality of their draft:

  • Following the advice of the Reviewer, a native English language professional has revised and edited the manuscript. In addition, the title has been modified to read: “Development of the “Applied Proteomics” concept for biotechnology applications in microalgae: example of the proteome data in “Nannochloropsis gaditana.”

More specific comments:

Very often, proteomic approaches lead to the identification of only a few peptides per protein with limited coverage. In most microalgae, which also have uncomplete genome annotations (if any), identifying proteins with confidence thus represents a real challenge. Wouldn’t that represent a major drawback for the proposed applied proteomics approach ?

  • We thank the reviewer for the comment. Of course, very often, proteomics approaches lead to the identification of only a few peptides per protein with limited coverage. This is the case in most microalgae (not model organisms), which also have incomplete genome annotations, if any. It is true therefore that identifying proteins with confidence represents a real challenge.

The case of a microorganism without a complete phylogenetic genome reference, it is more complicated to identify the proteins present by means of MS/MS. In particular, we had this problem with Pyrocystis lunula. We generated our own reference base, and the current tools allowed these processes to be carried out in an acceptable period of time. Once the proteome assay is done, the applied proteomics technique using the bioinformatic algorithm will be the same process as that described in N. gaditana.

Fajardo, C.; Amil-Ruiz, F.; Fuentes-Almagro, C.; De Donato, M.; Martinez-Rodriguez, G.; Escobar-Niño, A.; Carrasco, R.; Mancera, J.M.; Fernandez-Acero, F.J. An “omic” approach to Pyrocystis lunula: New insights related with this bioluminescent dinoflagellate. J. Proteomics 2019, 209, 103502, doi:10.1016/J.JPROT.2019.103502.

Lines 133-143 : This part about CRISPR/Cas9 genome editing is nice and all, but isn’t it a little bit out of topic ? Or at least, genome editing applications in the « applied proteomics » context should be more explained : could the CRISPR/Cas9 system help overproducing proteins of interest, delete some potentially problematic proteases ?....

  • Following the instructions of reviewers 1 and 2 the references to the CRIPR/Cas9 System have been deleted in the main text, because it is considered outside the paper’s topic.

Line 254 : « The applied proteomics have been transformed the proteomics data of N. gaditana in a high value industrial information [21]. ». This is the key part and yet, the underlying concept is not explained at all. How does the algorithm identifying proteins of biotechnological interest functions? Is it purely based on sequence alignments, on potential domains annotations ? On GO terms, biased towards biomedical applications ? And if in the Nannochloropsis dataset more than 400 proteins with potential industrial applications could be found, how to narrow down the list to candidates of interest ? Natural abundance should be a criterion I imagine.

  • We thank the reviewer for this comment. The algorithm is explained in the main text, following the reviewer’s suggestions: “The algorithm is based on sequence and semantic homology, on potential domains annotations and on comparison with other data sources such as United States patents [21].” This bio-algorithm was developed in experimentation by our laboratory with other companies during the national project collaboration. This algorithm is a valuable industrial secret for the company participating in the project, and is in the process of being patented. Because of this there cannot be a more detailed explanation of the bio-algorithm in the main text.

Lines 266-291 : This part denotes in the middle of the review… It looks like a mixture of Materials and Methods with a little bit of a Results section. Since the reported cloning, recombinant expression and antitumoral tests were previously reported in another reference [21] (Carrasco-Reinado R. et al.), the main essential finding that UCA01 was identified through the « applied proteomics » pipeline and has antitumoral activity should be reported in a more condensed paragraph.

  • Following the reviewer’s suggestion, this part has been modified in accordance with the reviewer’s advice, thus reducing the excessive details in the main text.

Reviewer 3 Report

  1. Minor suggestion were include throughout the PDF manuscript
  2. Figure 2 most be improved 
  3. Figure 3 most be improved

Author Response

Reviewer 3:

Comments and Suggestions for Authors

Minor suggestions were included throughout the PDF manuscript.

Check spelling

  • Following the advice of the Reviewer, the manuscript has been revised by a native English speaker. Also, the title has been modified.

Figure 2 must be improved. 

  • Following the reviewer’s suggestion, Figure 2 has been improved.

Figure 3 must be improved.

  • Following the reviewer’s suggestion, Figure 3 has been improved.